# PiRank: Scalable Learning To Rank via Differentiable Sorting

**Robin Swezey**[1*]      **Aditya Grover**[2,3]      **Bruno Charron**[1*]      **Stefano Ermon**[4]

[1]Amazon
[2]University of California, Los Angeles
[3]Facebook AI Research
[4]Stanford University
{rswezey, bcharron}@acm.org, adityag@cs.ucla.edu, ermon@cs.stanford.edu

## Abstract

A key challenge with machine learning approaches for ranking is the gap between the performance metrics of interest and the surrogate loss functions that can be optimized with gradient-based methods. This gap arises because ranking metrics typically involve a sorting operation which is not differentiable w.r.t. the model parameters. Prior works have proposed surrogates that are loosely related to ranking metrics or simple smoothed versions thereof, and often fail to scale to real-world applications. We propose PiRank, a new class of differentiable surrogates for ranking, which employ a continuous, temperature-controlled relaxation to the sorting operator based on NeuralSort [1]. We show that PiRank exactly recovers the desired metrics in the limit of zero temperature and further propose a divide-and-conquer extension that scales favorably to large list sizes, both in theory and practice. Empirically, we demonstrate the role of larger list sizes during training and show that PiRank significantly improves over comparable approaches on publicly available internet-scale learning-to-rank benchmarks.

## 1  Introduction

The goal of Learning-To-Rank (LTR) models is to rank a set of candidate items for any given search query according to a preference criterion [2]. The preference over items is specified via relevance labels for each candidate. The fundamental difficulty in LTR is that the downstream metrics of interest such as normalized discounted cumulative gain (NDCG) and average relevance position (ARP) depend on the ranks induced by the model. These ranks are not differentiable with respect to the model parameters, so the metrics cannot be optimized directly via gradient-based methods.

To resolve the above challenge, a popular class of LTR approaches map items to real-valued scores and then define surrogate loss functions that operate directly on these scores. Surrogate loss functions, in turn, can belong to one of three types. LTR models optimized via *pointwise* surrogates [3–6] cast ranking as a regression/classification problem, wherein the labels of items are given by their individual relevance labels. Such approaches do not directly account for any inter-dependencies across item rankings. *Pairwise* surrogate losses [7–14] can be decomposed into terms that involve scores of pairs of items in a list and their relative ordering. Finally, *listwise* surrogate losses [15–19] are defined with respect to scores for an entire ranked list. For many prior surrogate losses, especially those used for listwise approaches, the functional form is inspired via downstream ranking metrics, such as NDCG. However, the connection is loose or heuristically driven. For instance, SoftRank [14,19] introduces a Gaussian distribution over scores, which in turn defines a distribution over ranks and the surrogate is the expected NDCG w.r.t. this rank distribution.

---

[*]This work was done prior to joining Amazon.

35th Conference on Neural Information Processing Systems (NeurIPS 2021).

We propose PiRank, a listwise approach where the scores are learned via deep neural networks and the surrogate loss is obtained via a differentiable relaxation to the sorting operator. In particular, we choose as building block the temperature-controlled NeuralSort [1] relaxation for sorting and specialize it for commonly used ranking metrics such as NDCG and ARP. The resulting training objective for PiRank reduces to the exact ranking metric optimization in the limit of zero temperature and trades off bias for lower variance in the gradient estimates when the temperature is high. Furthermore, PiRank scales to real-world industrial scenarios where the size of the item lists is very large but the ranking metrics of interest are determined by only a small set of top ranked items. Scaling is enabled by a novel divide-and-conquer strategy akin to merge sort, where we recursively apply the sorting relaxation to sub-lists of smaller size and propagate only the top items from each sub-list for further sorting.

Empirically, we benchmark PiRank against 5 competing methods on two of the largest publicly available LTR datasets: MSLR-WEB30K [20] and Yahoo! C14. We find that PiRank is superior or competitive on 13 out of 16 ranking metrics and their variants, including 9 on which it is significantly superior to all baselines, and that it is able to scale to very large item lists. We also provide several ablation experiments to understand the impact of various factors on performance. To the best of our knowledge, this work is the first to analyze the importance of training list size on an LTR benchmark. Finally, we provide an open-source implementation[2] based on TensorFlow Ranking [21].

## 2    Background and Related Work

The LTR setting considers a finite dataset consisting of $n$ triplets $D = \{q_i, \{\boldsymbol{x}_{i,j}\}_{j=1}^L, \{y_{i,j}\}_{j=1}^L\}_{i=1}^n$. The $i$-th triplet consists of a query $q_i \in \mathcal{Q}$, a list of $L$ candidate items represented as feature vectors $\boldsymbol{x}_{i,j} \in \mathcal{X}$, and query-specific relevance labels $y_{i,j}$ for each item $j$. The relevance labels $y_{i,j}$ can be binary, ordinal or real-valued for more fine-grained relevance. For generality, we focus on the real-valued setting. Given a training dataset $D$, our goal is to learn a mapping from queries and itemsets to rankings. A ranking $\pi$ is a list of unique indices from $\{1, 2, \ldots, L\}$, or equivalently a permutation, such that $\pi_j$ is the index of the item ranked in $j$-th position. Without loss of generality, we assume lower ranks (starting from 1) have higher relevance scores. This is typically achieved by learning a scoring function $f : \mathcal{Q} \times \mathcal{X}^L \to \mathbb{R}^L$ that maps a query context and list of candidate items to $L$ scores. At test time, the candidate items are ranked by sorting their predicted scores in descending order. The training of $f$ itself can be done by a suitable differentiable surrogate objective, which we discuss next.

### 2.1    Surrogate Objectives for LTR

In this section, we briefly summarize prominent LTR approaches with a representative loss function for each category of pointwise, pairwise or listwise surrogate losses. We refer the reader to the excellent survey by [22] for a more extensive review. Omitting the triplet index, we denote the relevance labels vector as $\boldsymbol{y} \in \mathbb{R}^L$ and an LTR model's score vector obtained via the scoring function $f$ as $\hat{\boldsymbol{y}} \in \mathbb{R}^L$.

The simplest pointwise surrogate loss for ranking is the mean-squared error (MSE) between $\boldsymbol{y}$ and $\hat{\boldsymbol{y}}$:

$$\hat{\ell}_{\text{MSE}}(\boldsymbol{y}, \hat{\boldsymbol{y}}) = \frac{1}{L} \sum_{i=1}^L (\hat{y}_i - y_i)^2. \tag{1}$$

As the example loss above shows, pointwise LTR approaches convert ranking into a regression problem over the relevance labels and do not account for the relationships between the candidate items. Pairwise approaches seek to remedy this by considering loss terms depending in the predicted scores of pairs of items. For example, the widely used RankNet [9] aims to minimize the number of inversions, or incorrect relative orderings between pairs of items in the predicted ranking. It does so by modeling the probability $\hat{p}_{i,i'}$ that the relevance of the $i$-th item is higher than that of the $i'$-th item as a logistic map of their score difference, for all candidate items $i, i'$. The objective is then the

---

[2]https://github.com/ermongroup/pirank

cross-entropy:

$$\hat{\ell}_{\text{RankNet}}(\boldsymbol{y}, \hat{\boldsymbol{y}}) = -\sum_{i=1}^{L}\sum_{i'=1}^{L} \mathbb{1}\left(y_i > y_{i'}\right) \log \hat{p}_{i,i'} \tag{2}$$

where $\mathbb{1}(\cdot)$ denotes the indicator function and $\hat{p}_{i,i'}$ is a function of $\hat{\boldsymbol{y}}$. Pairwise approaches effectively model relationships between pairs of items and generally perform better than pointwise approaches, but still manifest limitations on downstream metrics which consider rankings in the full list and not just pairs. In fact, the larger the list of candidate items, the weaker these approaches tend to be: an error between the first and second item on a list is weighted the same in the RankNet loss as one between the last two items, despite the top items being of more importance in the LTR setting.

Listwise approaches learn from errors on the complete list. LambdaRank [13] extends RankNet by assigning weights to every loss term from Eq. 2:

$$\hat{\ell}_{\text{LambdaRank}}(\boldsymbol{y}, \hat{\boldsymbol{y}}) = -\sum_{i=1}^{L}\sum_{i'=1}^{L} \Delta\ell_{\text{NDCG}}(i, i') \log \hat{p}_{i,i'} \tag{3}$$

with $\Delta\ell_{\text{NDCG}}(i, i')$ the difference in the downstream metric NDCG (defined below) when swapping items $i$ and $i'$.

## 2.2 Ranking Metrics

Downstream metrics operate directly on the predicted ranking $\hat{\pi}$ (obtained by sorting $\hat{\boldsymbol{y}}$ in descending order) and the true relevance labels $\boldsymbol{y}$. They differ from conventional metrics used for other supervised learning problems in explicitly weighting the loss for each item by a suitably choosen increasing function of its predicted rank. For example, relevance position (RP) [23] multiplies the relevance labels with linearly increasing weights, and normalizes by the total relevance score for the query:

$$\text{RP}(\boldsymbol{y}, \hat{\pi}) = \frac{\sum_{j=1}^{L} y_{\hat{\pi}_j} j}{\sum_{j=1}^{L} y_j} \tag{4}$$

Averaging RP across the predictions made for all the queries in the test set gives the average relevance position (ARP) metric. Lower ARP signifies better performance.

A very common metric is the discounted cumulative gain (DCG) [24]. DCG computes the rescaled relevance of the $j$-th candidate by exponentiating its relevance label, and further divides it by the assigned log-ranking. This model incentivizes ranking models to focus on elements with higher graded relevance scores:

$$\text{DCG}(\boldsymbol{y}, \hat{\pi}) = \sum_{j=1}^{L} \frac{2^{y_{\hat{\pi}_j}} - 1}{\log_2(1 + j)} \tag{5}$$

A more common variant NDCG normalizes DCG by the maximum possible DCG attained via the optimal ranking $\pi^*$ (obtained by sorting $\boldsymbol{y}$ in descending order):

$$\text{NDCG}(\boldsymbol{y}, \hat{\pi}) = \frac{\text{DCG}(\boldsymbol{y}, \hat{\pi})}{\text{DCG}\left(\boldsymbol{y}, \pi^*\right)} \tag{6}$$

Higher DCG and NDCG signify better performance. Their truncated versions DCG@$k$ and NDCG@$k$ are defined by replacing $L$ with a cutoff $k$ in Eq. 5 so metrics are computed on the top-$k$ items.

## 3 Scalable and Differentiable Top-$k$ Ranking via PiRank

In PiRank, we seek to design a new class of surrogate objectives for ranking that address two key challenges with current LTR approaches. The first challenge is the gap between the downstream ranking metric of interest (e.g., NDCG, ARP) that involve a non-differentiable sorting operator and the differentiable surrogate function being optimized. The second challenge concerns the scalability w.r.t. the size of the candidate list $L$ for each query item. Larger list sizes are standard

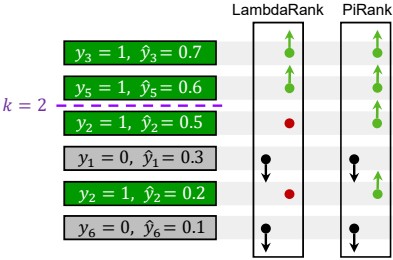

Figure 1: A set of items, green if relevant and gray otherwise, sorted by their score. Arrows show the sign of the loss derivative w.r.t. each item's predicted score for different methods (positive for black, negative for green and zero for red dots). Pairwise approaches weighted by differences in truncated ranking metrics, such as LambdaRank with NDCG@$k$, would put zero weights on the relevant items ranked below $k = 2$, thus bypassing learning signal. In comparison, PiRank efficiently learns from all items even using a $k = 2$ truncated loss.

in industrial applications but present computational and memory challenges for current approaches during both training and test-time inference. Pairwise and listwise methods (or hybrids) typically scale quadratically in the list size $L$, the number of items to rank for each query. Combining surrogates for truncated metrics, such as LambdaRank in Eq. 3 with NDCG@$k$ has a reduced complexity of $O(kL)$ but comes at the cost of vanishing gradient signal from relevant entries below $k$ (see Figure 1 for an illustration). Soft versions of the truncation metrics, such as Approximate NDCG Loss [25], can learn from all items but again scale quadratically with $L$ or do not take advantage of GPU acceleration [26].

As defined previously, a ranking $\pi$ is a list of indices equivalent to a permutation of $\{1, 2, \ldots, L\}$. The set of possible rankings can thus be seen as the symmetric group $\mathcal{S}_L$, of size $L!$. Every permutation $\pi$ can be equivalently represented as a permutation matrix $P_\pi$, an $L \times L$ matrix such that its $(i, \pi_i)$-th entry is 1 for all $i \in \{1, 2, \cdots, L\}$ and the remaining entries are all 0. We define the sorting operator $\mathrm{sort} : \mathbb{R}^L \to \mathcal{S}_L$ as a map of an $L$-dimensional input vector to the permutation that corresponds to the descending ordering of the vector components. Prior work in relaxing the sorting operator is based on relaxation of its output, either in the form of rankings [19, 26, 27], or permutation matrices [1, 28–30]. PiRank first applies the latter kind of relaxations to the ranking problem by introducing a new class of relaxed LTR metrics, then introduces a new relaxation that is particularly suited to these metrics.

### 3.1 Relaxed Ranking Metrics

We denote an LTR model by $f_\theta$ (e.g., deep neural network) with parameters $\theta$. The model outputs a vector of $L$ scores $\hat{\boldsymbol{y}} = f_\theta(q, \boldsymbol{x}_1, \ldots, \boldsymbol{x}_L)$ for a query $q$ and $L$ candidate elements $\{\boldsymbol{x}_i\}_{i=1}^L$. We first consider the NDCG target metric. In Eq. 6, the numerator $\mathrm{DCG}(\boldsymbol{y}, \hat{\pi})$ involves computing $\hat{\pi} = \mathrm{sort}(\hat{\boldsymbol{y}})$ which is non-differentiable w.r.t. $\theta$. Let $\boldsymbol{g}$ denote the column vector of graded relevance scores such that $g_j = 2^{y_j} - 1$. We can then rewrite $\mathrm{DCG}(\boldsymbol{y}, \hat{\pi})$ as:

$$\mathrm{DCG}(\boldsymbol{y}, \hat{\pi}) = \sum_{j=1}^{L} \frac{g_{\hat{\pi}_j}}{\log_2(1+j)} = \sum_{j=1}^{L} \frac{[P_{\hat{\pi}} \boldsymbol{g}]_j}{\log_2(1+j)}. \tag{7}$$

To obtain the DCG@$k$ objective, one can replace $L$ with $k$ in the sum. We omit the suffix @$k$ in the following, assuming that $k$ has been defined, potentially equal to $L$ which would yield the full metric.

Let $\widehat{P}_{\mathrm{sort}(\mathbf{s})}(\tau)$ denote a relaxation to the permutation matrix $P_{\mathrm{sort}(\mathbf{s})}$ that can be used for differentiable sorting of an input score vector $\mathbf{s}$, for some temperature parameter $\tau > 0$ such that the true matrix is recovered as $\tau \to 0^+$. Since $\hat{\pi} = \mathrm{sort}(\hat{\boldsymbol{y}})$, we can obtain a differentiable relaxation to $\mathrm{DCG}(\boldsymbol{y}, \hat{\pi})$:

$$\widehat{\mathrm{DCG}}(\boldsymbol{y}, \hat{\boldsymbol{y}}, \tau) = \sum_{j=1}^{k} \frac{[\widehat{P}_{\mathrm{sort}(\hat{\boldsymbol{y}})}(\tau) \boldsymbol{g}]_j}{\log_2(1+j)}. \tag{8}$$

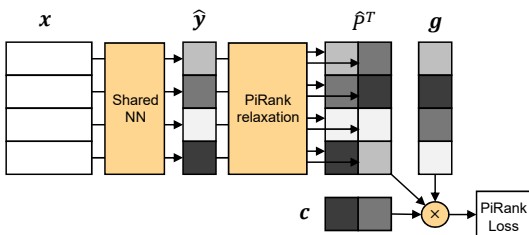

Figure 2: Architecture for the computation of the PiRank relaxed NDCG@$k$ loss for $L = 4$ and $k = 2$. Square cells represent scalars with darker shades indicating higher values. The fourth item has currently the highest score as given by the neural network but the second item has the highest relevance. The vector $\boldsymbol{c}$, with components $c_j = 1/\log(1 + j)$, discounts gains $\boldsymbol{g}$ based on rankings.

$$
\begin{pmatrix}
0 & \mathbf{0.9} & 0.1 \\
\mathbf{0.5} & 0.01 & 0.49 \\
\mathbf{0.5} & 0.09 & 0.41
\end{pmatrix}
\qquad
\begin{pmatrix}
\mathbf{0.8} & 0.2 & 0 \\
0.2 & 0.3 & \mathbf{0.5} \\
0.25 & \mathbf{0.6} & 0.15
\end{pmatrix}
$$

Figure 3: Doubly-stochastic (left) vs. unimodal (right) matrices. Maximum entry in every row in **bold**. Unlike unimodal matrices, two different items can have the same assignment of most-likely ranks (column indices) for doubly-stochastic matrix relaxations.

Substituting this in the expression for NDCG in Eq. 6, we obtain the following relaxation for NDCG:

$$
\widehat{\mathrm{NDCG}}(\boldsymbol{y}, \hat{\boldsymbol{y}}, \tau) = \frac{\widehat{\mathrm{DCG}}(\boldsymbol{y}, \hat{\boldsymbol{y}}, \tau)}{\mathrm{DCG}(\boldsymbol{y}, \pi^*)} \tag{9}
$$

where the normalization in the denominator does not depend on $\theta$ and can be computed exactly via regular sorting. Finally, we define the PiRank surrogate loss for NDCG as follows:

$$
\ell_{\mathrm{PiRank-NDCG}} = 1 - \widehat{\mathrm{NDCG}}(\boldsymbol{y}, \hat{\boldsymbol{y}}, \tau) \tag{10}
$$

which is bounded between 0 and 1 as is NDCG, and whose difference with the actual $(1 - \mathrm{NDCG})$ gets negligible as $\tau \to 0^+$. Figure 2 illustrates the model architecture for the above objective. Similarly, we can derive a surrogate loss for the ARP metric in Eq. 4 as:

$$
\hat{\ell}_{\mathrm{PiRank-ARP}}(\boldsymbol{y}, \hat{\boldsymbol{y}}, \tau) = \frac{\sum_{j=1}^{k} [\widehat{P}_{\mathrm{sort}(\hat{\boldsymbol{y}})}(\tau)\boldsymbol{y}]_j j}{\sum_{j=1}^{k} y_j}. \tag{11}
$$

## 3.2 Example: Differentiability via NeuralSort

Typically, relaxations to permutation matrices consider the Birkhoff polytope of doubly stochastic matrices. A doubly-stochastic matrix is a square matrix with entries in $[0, 1]$ where every row and column sum to 1. In contrast, NeuralSort [1] is a recently proposed relaxation of permutation matrices in the space of *unimodal* row-stochastic matrices. A unimodal matrix is a square matrix with entries in $[0, 1]$ such that the entries in every row sum to 1 (i.e. row-stochastic), but additionally enforce the constraint that the maximizing entry in every row should have a unique column index. See Figure 3 for an example of each type. Note that a unimodal matrix is not necessarily doubly-stochastic and vice versa. Permutation matrices are both doubly-stochastic and unimodal.

In NeuralSort [1], a unimodal relaxation of the permutation matrix $P_{\mathrm{sort}(\hat{\boldsymbol{y}})}$ can be defined as follows. Let $A_{\hat{\boldsymbol{y}}}$ denote the matrix of absolute pairwise score differences with $i, j$-th entry given as $[A_{\boldsymbol{y}}]_{ij} = |\hat{y}_i - \hat{y}_j|$. Then, the $i$-th row of the relaxed permutation matrix is:

$$
\widehat{P}_{\mathrm{sort}(\hat{\boldsymbol{y}})}^{(NS)}(\tau)_{i,\cdot} = \mathrm{softmax}\left[((L + 1 - 2i)\hat{\boldsymbol{y}} - A_{\hat{\boldsymbol{y}}}\mathbb{1})/\tau\right] \tag{12}
$$

where $\mathbb{1}$ is the vector with all components equal to 1. Its unimodal property makes it particularly well-suited to extracting top-$k$ items because, as seen in Figure 3, taking the maximizing elements of the first $k$ rows yields exactly $k$ items but may yield less in the case of a doubly-stochastic relaxation. However, the complexity to obtain the top-$k$ rows in this formulation, even for $k$ as low as 1, is quadratic in $L$ as the full computation of $A_{\hat{\boldsymbol{y}}}\mathbb{1}$ is required for the softmax operation in Eq. 12. This is prohibitive when $L \gg k$, a common scenario, and motivates the introduction of a new relaxation with a more favorable complexity for top-$k$ ranking.

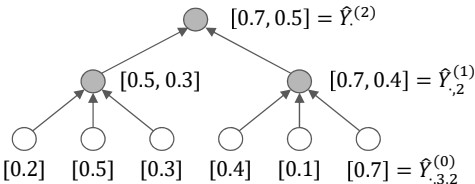

Figure 4: Divide-and-conquer strategy for $L = 6 = 3 \cdot 2$, $k = 2$ and $\hat{\boldsymbol{y}}^T = (0.2, 0.5, 0.3, 0.4, 0.1, 0.7)$. The scores are merged in groups of size $b_1 = 3$ and the respective top $k_1 = 2$ scores are kept, then the $b_2 = 2$ outputs are merged to obtain the final top $k_2 = k = 2$ scores. The effect of relaxation is not shown for readability. At non-zero temperature, the values at non-terminal nodes would be linear combination of the scores.

### 3.3  Scaling via Divide-And-Conquer

Our PiRank losses only require the first $k$ rows of the relaxed permutation matrix $\widehat{P}_{\text{sort}(\hat{\boldsymbol{y}})}$. This is specific to the LTR setting in which only the top-ranked items are of interest, in contrast to the full sorting problem that requires the full matrix. In PiRank, we leverage this insight to construct a divide-and-conquer variant of differentiable sorting relaxations such as NeuralSort to reduce the complexity of the metric computation. Our proposed construction can be viewed as a relaxed and truncated multi-way merge sort algorithm with differentiable sorting relaxations as building blocks. In the following discussion, we use NeuralSort as our running example while noting that the analysis extends more generally to other differentiable relaxations as well.

**Data Structure Construction.** Let $L = b_1 b_2 \cdots b_d$ be a factorization of the list size $L$ into $d$ positive integers. Using this factorization, we will construct a tree of depth $d$ with branching factor $b_j$ at height $j$. Next, we split the $L$-dimensional score vector $\hat{\boldsymbol{y}}$ into its $L$ constituent scalar values. We set these values as the leaves of the tree. See Figure 4 for an example. At every other level of the tree, we will merge values from the level below into equi-sized lists. Let $\{k_j\}_{j=0}^d$ be sizes for the intermediate results at level $j$, such that $k_0 = 1$ (leaves) and $\min(k, k_{j-1}b_j) \leq k_j \leq k_{j-1}b_j$ for $j \geq 1$ (explained below). Then, in an iterative manner for levels $j = 1, \ldots, d$, the value of a node at height $j$ are the top-$k_j$ scores given by the application of the NeuralSort operator on the concatenation of the values of its children. With $k_d = k$, the root value thus obtained is a relaxation of the top-$k$ scores in $\hat{\boldsymbol{y}}$. The top-$k$ rows of the relaxed permutation matrix $\widehat{P}_{\text{sort}(\hat{\boldsymbol{y}})}$ yielding these scores are constructed by compounding the operations at each iteration.

**Computational Complexity.** The intuition behind the favorable scaling is as follows. At step $j$, NeuralSort is applied on blocks of size $k_{j-1}b_j$ as it merges $b_j$ sub-blocks of size $k_{j-1}$. Obtaining the full sorted list of scores would require to keep all intermediate scores during the process, i.e., $k_j^{(\text{max})} = b_1 \cdots b_j = k_{j-1}b_j$ for $j \geq 1$. In the last step, the NeuralSort operator is applied on a list of size $k_{d-1}b_d$, equal to $L$ in this case, so the overall complexity would be at least quadratic in $L$ as explained previously. However, since only the top-$k$ scores are desired, intermediate outputs can be truncated if larger than $k$. Full truncation corresponds to $k_j^{(\text{min})} = \min(k, k_{j-1}b_j)$. Any choice $k_j^{(\text{min})} \leq k_j \leq k_j^{(\text{max})}$ is acceptable to recover the top-$k$ scores, with larger values allowing more information to flow at the expense of a higher complexity. Choosing $b_j \approx L^{1/d}$ and $k_j$ minimal, the list sizes $b_j k_{j-1}$ on which NeuralSort is applied at each step can thus be of the order of $L^{1/d}k$, much smaller than $L$ in the $d > 1$ and $k \ll L$ scenario.

Formally, let $\tau_1, \tau_2, \ldots, \tau_d$ be the relaxation temperatures at each height, with $\tau_d = \tau$ and $\tau_j \leq \tau_{j+1}$ for $j \in \{1, \ldots, d-1\}$. Define the tensor $\hat{Y}^{(0)}$ by reshaping $\hat{\boldsymbol{y}}$ to shape $(k_0, b_1, b_2, \ldots, b_d)$, yielding components

$$\hat{Y}^{(0)}_{1, i_1, i_2, \ldots, i_d} = \hat{y}_{1 + \sum_{j=1}^d (i_j - 1) \prod_{l=1}^{j-1} b_l}, \tag{13}$$

with $i_j \in \{1, \ldots, b_j\}$ and the first index is always 1 as $k_0 = 1$. With the tree representation, the first tensor index is the position in the node value vector and the rest of the indices identify the node by the index of each branching starting from the root. For $j \in \{1, 2, \ldots, d\}$, recursively define the tensors $\hat{Q}^{(j)}$, $\hat{Y}^{(j)}$ and $\hat{P}^{(j)}$ of respective shapes $(k_j, k_{j-1}, b_j, \ldots, b_d)$, $(k_j, b_{j+1}, \ldots, b_d)$ and

| Loss / Metric | OPA | ARP | MRR | NDCG@1 | NDCG@3 | NDCG@5 | NDCG@10 | NDCG@15 |
|---|---|---|---|---|---|---|---|---|
| RankNet | 0.611494 | 46.746979 | 0.786148 | 0.331595 | 0.336593 | 0.346928 | 0.375944 | 0.398582 |
| LambdaRank | 0.618954 | 46.174503 | 0.798169 | 0.392150 | 0.396045 | 0.404275 | 0.425611 | 0.444942 |
| Softmax | 0.612626 | 46.557617 | 0.761750 | 0.331527 | 0.338999 | 0.353011 | 0.381717 | 0.405312 |
| Approx. NDCG | 0.630616 | 45.461678 | **0.814873** | **0.423497** | 0.409272 | 0.414501 | 0.434463 | 0.453627 |
| NeuralSort | **0.635468** | **44.966999** | 0.779865 | 0.373344 | 0.386647 | 0.402052 | 0.430580 | 0.452863 |
| PiRank-NDCG | 0.629763 | 45.394020 | **0.813016** | **0.425006** | **0.420569** | **0.426034** | **0.446428** | **0.465309** |

| Loss / Metric | OPA | ARP | MRR | NDCG@1 | NDCG@3 | NDCG@5 | NDCG@10 | NDCG@15 |
|---|---|---|---|---|---|---|---|---|
| RankNet | 0.625170 | 10.514806 | 0.889641 | 0.599739 | 0.622155 | 0.650367 | 0.704332 | 0.733015 |
| LambdaRank | 0.636126 | 10.451870 | 0.896020 | 0.633181 | 0.652220 | 0.676243 | 0.723489 | 0.749001 |
| Softmax | 0.627313 | 10.472106 | 0.886976 | 0.588957 | 0.618409 | 0.648358 | 0.703746 | 0.732624 |
| Approx. NDCG | 0.648793 | 10.277598 | **0.903356** | **0.668700** | **0.670107** | 0.690353 | 0.735641 | 0.760539 |
| NeuralSort | 0.648416 | 10.320462 | 0.898246 | 0.640018 | 0.655990 | 0.681075 | 0.729225 | 0.754867 |
| PiRank-NDCG | **0.654255** | **10.250481** | 0.902088 | 0.661661 | **0.672390** | **0.693825** | **0.738479** | **0.763864** |

Table 1: Benchmark evaluation on (upper) MSLR-WEB30K and (lower) Yahoo! C14 test sets. In bold, the best performing method and all other methods not significantly worse.

$(k_j, b_1, \ldots, b_d)$ with components

$$\hat{Q}^{(j)}_{l,m,i_j,\ldots,i_d} = \text{softmax}\left[\left((k_{j-1}b_j + 1 - 2l)\hat{Y}^{(j-1)}_{m,i_j,\ldots,i_d} - \sum_{p=1}^{k_{j-1}}\sum_{q=1}^{b_j}\left|\hat{Y}^{(j-1)}_{m,i_j,i_{j+1},\ldots,i_d} - \hat{Y}^{(j-1)}_{p,q,i_{j+1},\ldots,i_d}\right|\right)/\tau_j\right],$$

(14)

$$\hat{Y}^{(j)}_{l,i_{j+1},\ldots,i_d} = \sum_{p=1}^{k_{j-1}}\sum_{q=1}^{b_j}\hat{Q}^{(j)}_{l,p,q,i_{j+1},\ldots,i_d}\hat{Y}^{(j-1)}_{p,q,i_{j+1},\ldots,i_d},$$

(15)

$$\hat{P}^{(j)}_{l,i_1,\ldots,i_d} = \sum_{m=1}^{k_{j-1}}\hat{Q}^{(j)}_{l,m,i_j,\ldots,i_d}\hat{P}^{(j-1)}_{m,i_1,i_2,\ldots,i_d},$$

(16)

with $\hat{P}^{(0)} = 1$. Intuitively, $\hat{Y}^{(j)}$ holds the relaxed top-$k_j$ scores at height $j$ and $\hat{Y}^{(d)}$ is the desired top-$k$ score vector. The interpretation of the indices in the tree structure is as for $\hat{Y}^{(0)}$, illustrated in Figure 4. More importantly, we keep track of the relaxed sorting operation that yielded this output. $\hat{Q}^{(j)}$ is the relaxed permutation matrix obtained by applying NeuralSort in Eq. 12 to $\hat{Y}^{(j)}$, while $\hat{P}^{(j)}$ compounds the relaxed permutation matrices obtained so far so it always maps from the initial list size. Finally, define the $k \times L$ matrix $\hat{P}$ by reshaping the tensor $\hat{P}^{(d)}$, yielding components

$$\hat{P}_{l,1+\sum_{j=1}^{d}(i_j-1)\prod_{l=1}^{j-1} b_l} = \hat{P}^{(d)}_{l,i_1,\ldots,i_d},$$

(17)

for $i_j \in \{1, \ldots, b_j\}$. The $k$ rows of $\hat{P}$ are used as the top-$k$ rows of the relaxed sorting operator $\widehat{P}_{\text{sort}(\hat{y})}(\tau)$. This approach is equivalent to NeuralSort, yielding Eq. 12 for $d = 1$. Proof of convergence for $\tau \to 0^+$ of this relaxation in the general case $d \geq 1$ is presented in Appendix B.

In the simple case where $L = b^d$ and we set $b_j = b$, $k_j = \min(k, b^j)$ for all $j \in \{1, \ldots, d\}$, the complexity to compute $\hat{P}$ and thus the PiRank losses is then $O(L^{1+1/d} + (d-1)k^2 L)$, which scales favorably in $L$ if $d > 1$ and $k = O(1)$. In the general case, the score list can be padded, e.g. to the power of 2 following $L$, such that the previous complexity holds for $b = 2$ and $d = \lceil \log_2 L \rceil$, but other factorizations may yield lower complexity depending on $L$.

## 4 Experiments

We present two sets of experiments in this section: (a) benchmark evaluation comparing PiRank with other ranking based approaches on publicly available large-scale benchmark LTR datasets, and (b) ablation experiments for the design choices in PiRank.

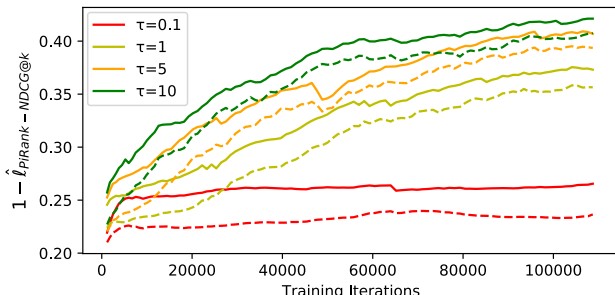

Figure 5: $1 - \hat{\ell}_{\mathrm{PiRank-NDCG@k}}$ ($k = 10$, full lines) for different values of the temperature parameter $\tau$, with the corresponding value of the hard metric NDCG@10 (dashed lines), at validation.

## 4.1 Benchmark Evaluation via TF-Ranking

**Datasets.** To empirically test PiRank, we consider two of the largest open-source benchmarks for LTR: the MSLR-WEB30K[3] and the Yahoo! LTR dataset C14[4]. Both datasets have relevance scores on a 5-point scale of 0 to 4, with 0 denoting complete irrelevance and 4 denoting perfect relevance. We give extensive details on the datasets and experimental protocol in Appendix C.

**Baselines.** We focus on neural network-based approaches and use the open-source TensorFlow Ranking (TFR) framework [21]. TFR consists of high-quality GPU-friendly implementations of several LTR approaches, common evaluation metrics, and standard data loading formats. We compare our proposed loss, PiRank-NCDG, with the following baselines provided by TensorFlow Ranking: Approximate NDCG Loss [25], Pairwise Logistic Loss (RankNet, Eq. 2), Pairwise Logistic Loss with lambda-NDCG weights (LambdaRank, Eq. 3), and the Softmax Loss. We also include NeuralSort, whose loss is the cross-entropy of the predicted permutation matrix. Of these methods, the Pairwise Logistic Loss (RankNet) is a pairwise approach while the others are listwise. While our scope is on differentiable ranking surrogate losses for training neural networks, other methods such as the tree-based LambdaMART [31] could potentially yield better results.

**Setup.** All approaches use the same 3-layer fully connected network architecture with ReLU activations to compute the scores $\hat{y}$ for all (query, item) pairs, trained on 100,000 iterations. The maximum list size for each group of items to score and rank is fixed to 200, for both training and testing. Further experimental details are deferred to Appendix C. We evaluate Ordered Pair Accuracy (OPA), Average Relevance Position (ARP), Mean Reciprocal Rank (MRR), and NDCG@$k$ with $k \in \{1, 3, 5, 10, 15\}$. We determine significance similarly to [32]. For each metric and dataset, the best performing method is determined, then a one-sided paired t-test at a 95% significance level is performed on query-level metrics on the test set to compare the best method with every other method.

**Results.** Our results are shown in Table 1. Overall, PiRank shows similar or better performance than the baselines on 13 out of 16 metrics. It significantly outperform all baselines on NDCG@$k$ for $k \geq 5$ while Approx. NDCG is competitive on NDCG@$k$ for $k \leq 3$ and MRR. PiRank is also significantly superior on OPA and ARP metrics on Yahoo! C14 while NeuralSort is superior for MSLR-WEB30K.

## 4.2 Ablation Experiments

**Temperature.** The temperature hyperparameter $\tau$ is used in PiRank to control the degree of relaxation. We experiment on several values ($\tau \in \{0.1, 1, 5, 10\}$) using the MSLR-WEB30K dataset and the experimental settings for ablation provided in Appendix C. Figure 5 demonstrates the importance of correctly tuning $\tau$. High values ($\tau > 1$) speed up training, especially in the early regime, while low values induce large gradient norms which are unsuitable for training and lead to the loss stalling or even diverging. Another observation is that the relaxed metric $1 - \hat{\ell}_{\mathrm{PiRank-NDCG@k}}$

---

[3]https://www.microsoft.com/en-us/research/project/mslr/
[4]https://webscope.sandbox.yahoo.com

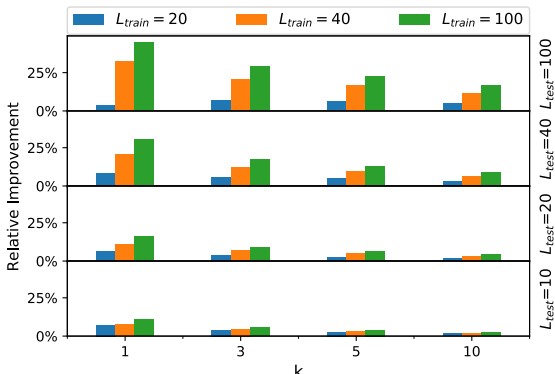

Figure 6: Relative improvement of NDCG@$k$ on different values of $L_{test}$, for different $L_{train}$ values vs. a baseline of $L_{train} = 10$.

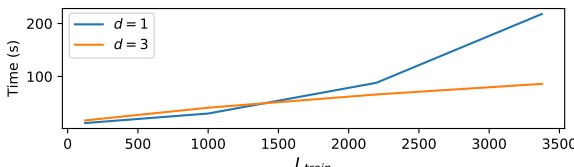

Figure 7: Wall-clock time for 100 training steps, each corresponding to 16 queries, for different $L_{train}$ and maximal depth $d$. We use $k = 1$ and $L_{train} = 5^3, 10^3, 13^3$ and $15^3$ such that $L_{train} = b^3$ for $d = 3$ and $k_j = 1$. Variation across runs is too small ($\sim$1s) and omitted for brevity.

closely follows the value of the downstream metric NDCG@$k$ as expected. We further experimented with a very high temperature value and an exponentially decreasing annealing schedule in Appendix D.

**Training List Size.** The training list size parameter $L_{train}$ determines the number of items to rank for a query during training. The setting is the same as for the temperature ablation experiment, but with training list sizes $L_{train} \in \{10, 20, 40, 100\}$ which we then evaluate on testing list sizes in the same range of values $L_{test} \in \{10, 20, 40, 100\}$. The dataset is again MSLR-WEB30K. Figure 6 exposes four patterns for NDCG@$k$. First, for a fixed $L_{test}$ and $k$, a larger $L_{train}$ is always better. Second, for a fixed $L_{test}$, we observe diminishing returns along $k$, as relative improvements decrease for all $L_{train}$. This observation is confounded by NDCG@$k$ values growing larger with $k$, but the metric is always able to distinguish between ranking functions [33]. Third, for a fixed $k$, our returns along $L_{test}$ increase with $L_{train}$ (except for $L_{train} = 20$ and $k = 1$). This means that the need for a larger $L_{train}$ is more pronounced for larger values of $L_{test}$. Fourth and last, the returns increase most dramatically with $L_{train}$ when $L_{test} \gg k$ (top left), a common industrial setting. Values for NDCG@$k$, MRR, OPA, ARP are provided in Appendix D. For MRR, using a larger $L_{train}$ is always beneficial regardless of $L_{test}$, but not always for OPA and ARP.

**Depth.** A main advantage of the PiRank loss is how it can scale to very large training list sizes $L_{train}$. This setting is difficult to come across with traditional LTR datasets, which are manually annotated, but occurs frequently in practice. One example is when the relevance labels are obtained from implicit signals such as clicks or purchases in recommendation systems. In this case, an LTR model is used to re-rank a list of candidates generated by another, simpler, model choosing among all possible items those potentially relevant to a query or context. An LTR model capable of handling very large lists can reduce the impact of errors made by the simpler candidate generation step, moving to the top an item lowly ranked at first that would have been cut off from a smaller list. To test the complexity shown in Section 3.2 in extreme conditions, we create a synthetic dataset as described in Appendix E. Figure 7 shows how the training time for depth $d = 3$ scales much more favorably than for $d = 1$, following their respective time complexities of $O(L^{1+1/3})$ and $O(L^2)$.

## 5  Summary and Limitations

We proposed PiRank, a novel class of surrogate loss functions for Learning-To-Rank (LTR) that leverages a continuous, temperature-controlled relaxation to the sorting operator [1] as a building block. This allows us to recover exact expressions of the commonly used non-differentiable ranking metrics in the limit of zero temperature, which we proved in particular for the NDCG metric. Crucially, we proposed a construction inspired by the merge-sort algorithm that permits PiRank to scale to very large lists.

In our experiments on the largest publicly available LTR datasets, we observed that PiRank has superior or similar performance with competing methods on the MSLR-WEB30K and Yahoo! C14 benchmarks on 13/16 ranking metrics and their variants.

As future work, we would like to explore other recent relaxations of the sorting operator [26, 34, 35] as a building block for the PiRank framework. Further, as ranking is a core component of modern day technology and influences everyday decision making pipelines involving vulnerable populations, care needs to be taken that our proposed systems are extended to account for biases and fairness criteria when deployed in real world settings.

## 6  Acknowledgements

Robin Swezey and Bruno Charron were supported by Rakuten, Inc.
Stefano Ermon is supported in part from NSF (#1651565, #1522054, #1733686), ONR (N00014-19-1-2145), AFOSR (FA9550-19-1-0024) and Bloomberg.

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
