# Appendices

## A  Further Related Works

**Pairwise approaches.**  Closely related to RankNet, pairwise approaches such as Sortnet [36] and SmoothRank [27] casts sorting of $n$ elements as performing $n^2$ pairwise comparisons, and try to approximate the pairwise comparison operator for sorting. We consider a more direct relaxation with attractive properties for rankings that we describe in Section 3.

**Listwise approaches.**  include ListNet [37] and ListMLE [38], which define surrogate losses that take into consideration the full predicted rank ordering while being agnostic to the downstream ranking metrics. ListNet for instance considers the predicted scores as parameters for the Plackett-Luce distribution [39,40] and learns these scores via maximum likelihood estimation.

## B  Proof of Convergence

Used in a PiRank surrogate loss of Section 3.1, the relaxation presented in Section 3.2 recovers the downstream metric by lowering the temperature as formalized in the result below for NDCG.

**Proposition 1.** *If we assume that the entries of $\hat{\boldsymbol{y}}$ are drawn independently from a distribution that is absolutely continuous w.r.t. the Lebesgue measure in $\mathbb{R}$, then the following convergence holds almost surely:*

$$\lim_{\tau \to 0^+} \hat{\ell}_{\text{PiRank-NDCG}}(\boldsymbol{y}, \hat{\boldsymbol{y}}, \tau) = 1 - \text{NDCG}(\boldsymbol{y}, \hat{\pi}) \tag{18}$$

*where $\hat{\pi} = \text{sort}(\hat{\boldsymbol{y}})$.*

*Proof.* In the $d > 1$ case, the limit is interpreted as

$$\lim_{\tau \to 0^+} = \lim_{\tau_d \to 0^+} \lim_{\tau_{d-1} \to 0^+} \dots \lim_{\tau_1 \to 0^+} \tag{19}$$

given the increasing ordering of the temperatures by height and the constraint $\tau_d = \tau$.

We first sketch a proof by induction on the height $j$ that, under the same assumptions as the proposition, for all $i_{j+1}, \cdots, i_d$, the $k'_j$-dimensional vector

$$\boldsymbol{y}^{(j)}_{i_{j+1},\cdots,i_d} \equiv \lim_{\tau_j \to 0^+} \hat{Y}^{(j)}_{:k'_j,i_{j+1},\cdots,i_d} \tag{20}$$

with $k'_j = \min(k, k_j)$ and $:l$ the top-$l$ rows extraction, contains the top-$k'_j$ scores in $\hat{Y}^{(0)}_{\cdot,\cdots,\cdot,i_{j+1},\cdots,i_d}$ in descending order and the $k'_j \times L_j$ matrix

$$P^{(j)}_{i_{j+1},\cdots,i_d} \equiv \lim_{\tau_j \to 0^+} \hat{P}^{(j)}_{:k'_j,i_{j+1},\cdots,i_d} \tag{21}$$

with $L_j = b_1 \cdots b_j$ is the row-truncated permutation matrix realizing the ordering,

$$\boldsymbol{y}^{(j)}_{i_{j+1},\cdots,i_d} = P^{(j)}_{i_{j+1},\cdots,i_d} \hat{Y}^{(0)}_{\cdot,\cdots,\cdot,i_{j+1},\cdots,i_d} \tag{22}$$

where reshaping as necessary is implicit in the above two equations.

For $j = 0$, this is trivial as $P^{(0)} = 1$ and by convention $b_0 = k_0 = 1$.

Assuming the above is true for a height $j - 1$, the top-$k'_j$ scores in $\hat{Y}^{(0)}_{\cdot,\cdots,\cdot,i_{j+1},\cdots,i_d}$ are included in the concatenation of the vectors $\hat{Y}^{(j-1)}_{\cdot,i_j,\cdots,i_d}$ for $i_j \in \{1,\ldots,b_j\}$ in the $\tau_{j-1} \to 0^+$ limit from the assumption (no limit for $j = 1$). $\hat{Q}^{(j)}_{\cdot,\cdot,\cdot,i_{j+1},\cdots,i_d}$ is then the NeuralSort relaxed permutation matrix for these concatened vector. From Theorem 1 of [1], we know that in the $\tau_j \to 0^+$ limit, this matrix will converge to the sorting permutation matrix. In this limit, $\hat{Y}^{(j)}_{\cdot,i_{j+1},\cdots,i_d}$ is then sorted version of the concatened vector, so that in particular its top-$k'_j$ elements are the sorted top-$k'_j$ elements of the

Table 2: Shared parameter values for benchmark (Section 4.1) and ablation (Section 4.2) experiments.

| Parameter | Benchmark | Ablation |
|---|---|---|
| Hidden layer sizes | 1024,512,256 | 256,256,128,128,64,64 |
| Hidden layer activations | ReLu | ReLu |
| Batch normalization | Yes | No |
| Dropout rate | 0.3 | 0 |
| Batch size | 16 | 16 |
| Learning rate | 1.00E-03 | 1.00E-05 |
| Optimizer | Adam | Adam |
| Iterations | 100,000 steps | 100 epochs |
| Training list size $L_{train}$ | 200 | 100 when fixed |
| Testing list size $L_{test}$ | 200 | 100 when fixed |
| Temperature $\tau$ (PiRank & NeuralSort) | 1000 | 5 when fixed |
| Straight-through estimation (PiRank & NeuralSort) | Yes | Yes |
| NDCG cutoff $k$ (PiRank & LambdaRank) | 10 | 10 |
| Depth $d$ (PiRank) | 1 | 1 when fixed |

concatenated vector, proving the claim on $\boldsymbol{y}^{(j)}_{i_{j+1},\cdots,i_d}$. Further, the claim on $P^{(j)}$ directly derives from the previous observation on the limit of $\widehat{Q}^{(j)}_{\cdot,\cdot,\cdot,i_{j+1},\cdots,i_d}$ and the fact that a product of permutation matrices which is the matrix of the product of the permutations. This finishes the proof by induction.

Taking $j = d$, we obtain from Eq. 22 and the nature of permutation matrices that

$$\lim_{\tau \to 0^+} \widehat{P}_{\text{sort}(\hat{y})}(\tau)_{:k} = \left[ P_{\text{sort}(\hat{y})} \right]_{:k}. \tag{23}$$

From limit calculus, we know that the limit of finite sums is the sum of the limits and hence, substituting the above result in Eq. 8 we have:

$$\lim_{\tau \to 0^+} \widehat{\text{DCG}}(\boldsymbol{y}, \hat{\boldsymbol{y}}, \tau) = \text{DCG}(\boldsymbol{y}, \hat{\pi}). \tag{24}$$

Substituting the above in Eq. 9 and Eq. 10 proves the proposition.

$\square$

Note that the assumption of independent draws is needed to ensure that the elements of $\hat{\boldsymbol{y}}$ are distinct almost surely.

## C   Experimental Details

**Datasets.**   We test PiRank on MSLR-WEB30K[5] and the Yahoo! LTR dataset C14[6]. MSLR-WEB30K contains 30,000 queries from Bing with feature vectors of length 136, while Yahoo! C14 dataset comprises 36,000 queries, 883,000 items and feature vectors of length 700. In both datasets, the number of items per query can exceed 100, or even 1,000 in the case of MSLR-WEB30K. Both datasets have relevance scores on a 5-point scale of 0 to 4, with 0 denoting complete irrelevance and 4 denoting perfect relevance. Note that when using binary classification-based metrics such as mean-reciprocal rank, ordinal relevance score from 1 to 4 are mapped to ones. MSLR-WEB30K is provided in folds of training / validation / test sets rotating on 5 subsets of data, and we choose to use Fold1 for our experiments. For Yahoo! C14, we use "Set 1" which is the larger of the two provided sets. For both datasets, we use the standard train/validation/test splits. We use the validation split for both early stopping and hyperparameter selection for all approaches.

**TFR Implementation.**   We provide a TensorFlow Ranking implementation of the PiRank NDCG Loss as well as the original NeuralSort Permutation Loss which can be plugged in directly into TensorFlow Ranking.[7]

---

[5]https://www.microsoft.com/en-us/research/project/mslr/

[6]https://webscope.sandbox.yahoo.com

[7]https://github.com/ermongroup/pirank

Table 3: Training list size effectiveness on ranking metrics

| OPA | $L_{train}$ | | | |
|---|---|---|---|---|
| $L_{test}$ | 10 | 20 | 40 | 100 |
| 10 | 0.5830 | 0.5947 | 0.5939 | **0.5949** |
| 20 | 0.5852 | 0.5949 | **0.5961** | 0.5926 |
| 40 | 0.5816 | 0.5935 | **0.5942** | 0.5915 |
| 100 | 0.5755 | 0.5859 | **0.5867** | 0.5844 |

| MRR | $L_{train}$ | | | |
|---|---|---|---|---|
| $L_{test}$ | 10 | 20 | 40 | 100 |
| 10 | 0.6691 | 0.6830 | 0.6912 | **0.6949** |
| 20 | 0.6835 | 0.7048 | 0.7087 | **0.7172** |
| 40 | 0.6732 | 0.7042 | 0.7230 | **0.7350** |
| 100 | 0.6628 | 0.6985 | 0.7301 | **0.7548** |

| ARP | $L_{train}$ | | | |
|---|---|---|---|---|
| $L_{test}$ | 10 | 20 | 40 | 100 |
| 10 | 5.0164 | 4.9584 | 4.9662 | **4.9428** |
| 20 | 9.4277 | 9.3431 | **9.3334** | 9.3401 |
| 40 | 18.3042 | 18.0688 | **18.0493** | 18.0617 |
| 100 | 42.9107 | 42.4183 | **42.3972** | 42.4091 |

| NDCG@1 | $L_{train}$ | | | |
|---|---|---|---|---|
| $L_{test}$ | 10 | 20 | 40 | 100 |
| 10 | 0.3850 | 0.4127 | 0.4140 | **0.4261** |
| 20 | 0.3320 | 0.3521 | 0.3670 | **0.3860** |
| 40 | 0.2829 | 0.3054 | 0.3403 | **0.3683** |
| 100 | 0.2569 | 0.2665 | 0.3401 | **0.3713** |

| NDCG@3 | $L_{train}$ | | | |
|---|---|---|---|---|
| $L_{test}$ | 10 | 20 | 40 | 100 |
| 10 | 0.4610 | 0.4793 | 0.4826 | **0.4878** |
| 20 | 0.3757 | 0.3885 | 0.4017 | **0.4092** |
| 40 | 0.3188 | 0.3373 | 0.3572 | **0.3731** |
| 100 | 0.2780 | 0.2963 | 0.3349 | **0.3579** |

| NDCG@5 | $L_{train}$ | | | |
|---|---|---|---|---|
| $L_{test}$ | 10 | 20 | 40 | 100 |
| 10 | 0.5358 | 0.5498 | 0.5531 | **0.5570** |
| 20 | 0.4181 | 0.4271 | 0.4388 | **0.4441** |
| 40 | 0.3447 | 0.3607 | 0.3780 | **0.3896** |
| 100 | 0.2971 | 0.3158 | 0.3461 | **0.3635** |

| NDCG@10 | $L_{train}$ | | | |
|---|---|---|---|---|
| $L_{test}$ | 10 | 20 | 40 | 100 |
| 10 | 0.6994 | 0.7100 | 0.7115 | **0.7141** |
| 20 | 0.5090 | 0.5165 | 0.5257 | **0.5305** |
| 40 | 0.3989 | 0.4106 | 0.4243 | **0.4337** |
| 100 | 0.3330 | 0.3485 | 0.3720 | **0.3878** |

**Straight-through Estimation.** The PiRank surrogate learning objective can be optimized via two gradient-based techniques in practice. The default mode of learning is to use the relaxed objective during both forward pass for evaluating the loss and for computing gradients via backpropogation. Alternatively, we can perform *straight-through estimation* [41], where we use the hard version for evaluating the loss forward, but use the relaxed objective in Eq. 9 for gradient evaluation. We observe improvements from the latter option and use it throughout. The hard version can be obtained via exact sorting of the predicted scores. In the context of a unimodal relaxation (Sec 3.2), a hard version can also be obtained via a row-wise arg max operation of the relaxed permutation matrix, which recovers an actual permutation matrix usable in the downstream objective.

**Architecture and Parameters.** Experiment parameters that are shared across losses, such as the scoring neural network architecture, batch size, training and test list sizes, are provided in Table 2, along with loss-specific parameters if they differ from the default TensorFlow Ranking setting.

**Experimental Workflow.** We rely on TensorFlow Ranking for most of our work outside the NeuralSort and PiRank loss implementations, which takes care of query grouping, document list tensor construction, baseline implementation and metric computation among others.

**Computing infrastructure.** The experiments were run on a server with 4 8-core Intel Xeon E5-2620v4 CPUs, 128 GB of RAM and 4 NVIDIA Telsa K80 GPUs.

**Libraries and Software.** This work extensively relied on GNU Parallel [42] and the Sacred library [8] for experiments.

**Licenses.** TensorFlow Ranking is licensed under the Apache License 2.0 [9]. GNU Parallel is licensed under the GNU General Public License [10]. Sacred is licensed under the MIT License [11]. The dataset MSLR-WEB30K is licensed under the Microsoft Research License Agreement (MSR-LA). The license files for the dataset Yahoo! C14 are provided in the datasets at download time from their homepages, and included in the supplemental material. Our released PiRank code is licensed under the MIT license.

## D  Ablation Experiments

We provide all results for the temperature experiments described in Section 4.2, in Figures 8, 9, 10, 11, 12, 5. The smoothing parameter used in all plots is 0.9, the number of data points is 100 epochs for all figures except the training loss (1,000 iterations). We provide additional plots with a basic exponentially decreasing annealing schedule and a very high temperature value of 1e12 to show limits of the relaxation on Figure 13. Full results for the ablation experiments on the training list size described in Section 4.2 are provided in Table 3.

## E  Synthetic LTR Data

To the best of our knowledge, there is no public LTR dataset with very large numbers of documents per query ($L > 1000$). We thus propose the following synthetic dataset for testing and development at scale (see Section 4.2):

For each query $q_i$, $i \in \{1, \cdots, n\}$,

1. Generate $L$ documents $\{\mathbf{x}_{i,j}\}_{j=1}^{L}$ where $\mathbf{x}_{i,j}$ is a vector of $m_d$ $\phi$-distributed document features.

2. Randomly pick a vector $c_i$ of $m_q \leq m_d$ column indices from $\{1, \cdots, m_d\}$ without replacement.

3. Generate $\psi$-distributed query features $\{\gamma_i\}_{k=1}^{m_q}$.

4. Compute labels capped between $\ell$ and $h$ s.t.
   $y_{i,j} = max(\ell, min(h, \sum_{k=1}^{m_q} \gamma_k x_{i,j,c_k}))$.

5. Concatenate the query features $\{\gamma_i\}_{k=1}^{m_q}$ to each $\mathbf{x}_{i,j}$.

This process allows us to generate datasets of arbitrarily large size, where we control $L$, $n$, $m$, $c$ and the distributions $\phi$ and $\psi$. The process is easy to reuse, and made available in our TFR codebase.

---

[8] https://github.com/IDSIA/sacred

[9] https://github.com/tensorflow/ranking/blob/master/LICENSE

[10] https://www.gnu.org/licenses/gpl-3.0.html

[11] https://github.com/IDSIA/sacred/blob/master/LICENSE.txt

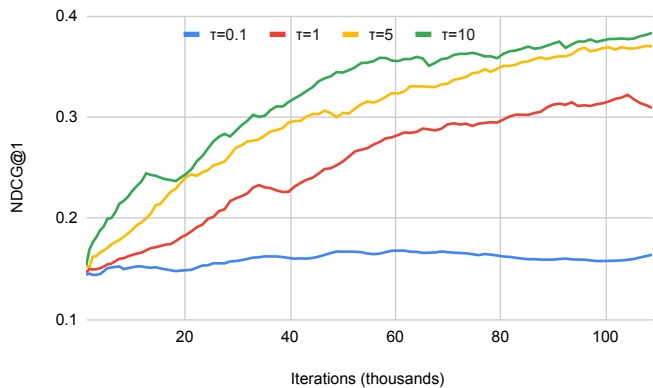

Figure 8: Validation NDCG@1 during PiRank training parametrized by temperature $\tau$

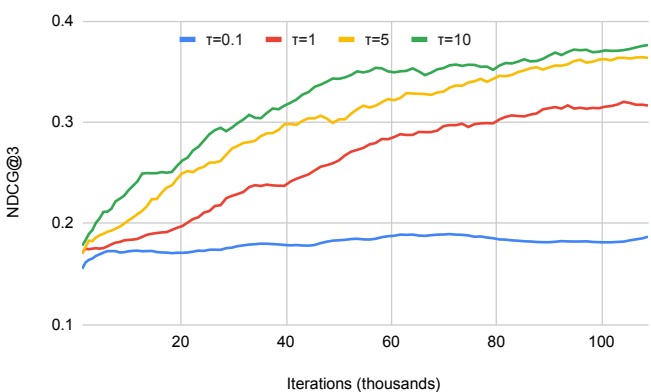

Figure 9: Validation NDCG@3 during PiRank training parametrized by temperature $\tau$

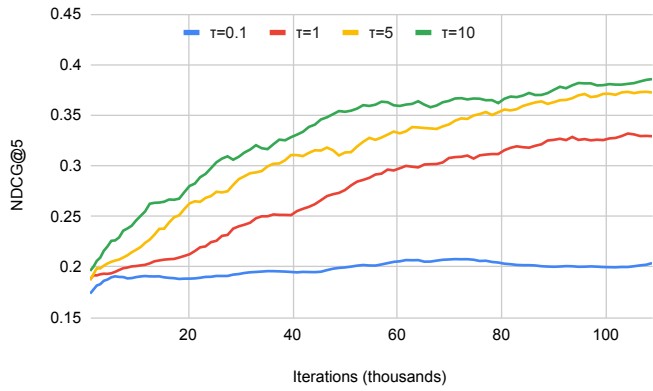

Figure 10: Validation NDCG@5 during PiRank training parametrized by temperature $\tau$

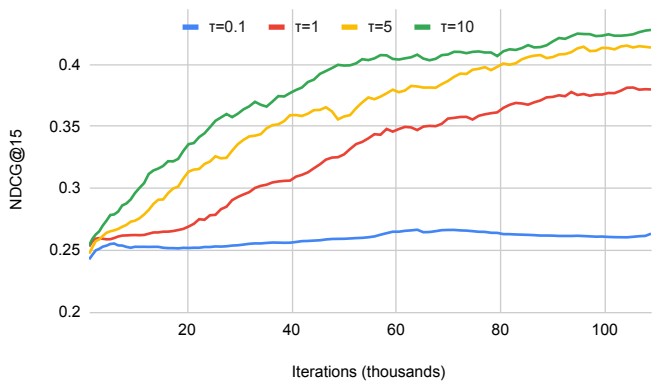

Figure 11: Validation NDCG@15 during PiRank training parametrized by temperature $\tau$

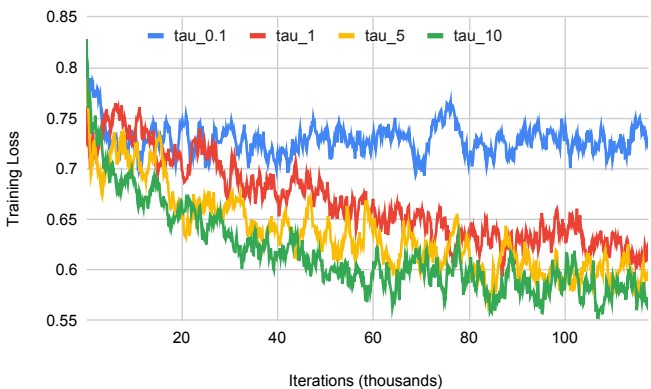

Figure 12: Training loss during PiRank training parametrized by temperature $\tau$

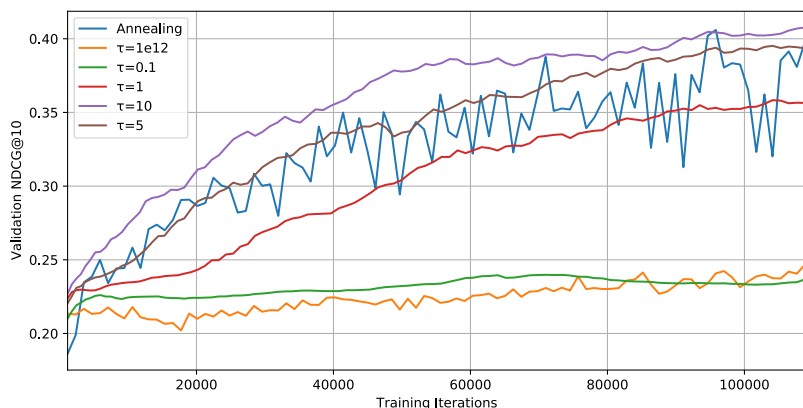

Figure 13: Validation NDCG@10 for PiRank-NDCG@10 using the experimental settings of Section 4.2. This figure shows the validation NDCG@10 from Figure 5 superimposed with an annealing schedule temperature (blue) and a very high temperature of 1e12 (orange).