# OpenReview forum: "PiRank: Scalable Learning To Rank via Differentiable Sorting"
_NeurIPS.cc/2021/Conference — NeurIPS 2021 Poster_

### Official Review · Reviewer_scsx · 2021-07-15

**Rating:** 6
**Confidence:** 3

**Summary:**

The authors provide a new scalable listwise loss for ranking called PiRank.
The loss is defined as 1 - a differentiable relaxation of NDCG metric with temperature-controlled inspired by NeuralSort algorithm.
In this setting, the difference with the NDCG is negligeable the temperature tends to 0.
Scalability is ensured by a divide-and-conquer strategy where the sorting relaxation is applied to sub-lists of smaller size and then propagate only on the top items from each sub-list to merge the sort.

**Ethical Concerns:**

Not applicable.

**Limitations And Societal Impact:**

Limitations:
- Can we really say PiRank is a listwise approach? The loss is still based on pairwise comparisons.
- It would be helpful for the reader to recall the main differences between the baselines and PiRank in the experiment session.

Societal impact: this is not discussed in the paper because it is not the main object. This is ranking based on relevance only not taking into account diversity, fairness etc.

**Main Review:**

In general, the paper is relatively clear and easy to follow.

The authors start by presenting the 3 representative loss function for pointwise, pairwise and listwise surrogate losses alongside the mainly used ranking metrics. The paper is really of interest of the community as it proposes a surrogate loss which approximately optimizes the main used ranking metric: NDCG.

The authors provide experiments on two large publicly available datasets with 18 different ranking metrics. The code will be made available which is highly appreciated for reproducibility of the results.

**Time Spent Reviewing:**

4

---

> ### Author Response · Authors · 2021-08-22
> **Response to reviewer's feedback**
>
> We thank the reviewer for their review and answer the outstanding comments and questions below.
>
> Q: Can we really say PiRank is a listwise approach? The loss is still based on pairwise comparisons.
>
> A: Pairwise losses can be decomposed into a sums of terms, where each term only involve the scores of a pair of documents, e.g. Eq (2). Listwise losses have no such restriction as can be seen in Eq (3) where ΔlNDCG(i,i’) involves all scores and not only those of documents i and i’. PiRank cannot be decomposed as a sum of pairwise terms and is thus a listwise approach.
>
> Q: It would be helpful for the reader to recall the main differences between the baselines and PiRank in the experiment session.
>
> A: We thank the reviewer for their suggestion and will do so in the final version.
>
> Again, we are very thankful for the reviewer’s helpful feedback and comments. Please let us know if there are any remaining questions; we will be glad to address them!

---

### Official Review · Reviewer_BM9C · 2021-07-15

**Rating:** 6
**Confidence:** 4

**Summary:**

This paper proposes PiRank, a new neural learning-to-rank (LTR) approach based on differentiable sorting operators. The authors present empirical results against two widely used LTR data sets (MSLR-WEB30K and Yahoo! C14).

**Limitations And Societal Impact:**

The authors adequately addressed the limitations and potential negative societal impact of their work.

**Main Review:**

This paper is generally well-written, flows logically, and is easy to read. I did not notice any major issues with the presentation. The proposed methods and experimental results are also clearly described.

The proposed algorithm, which is based on NeuralSort is interesting, technically sound, and novel.

The experimental evaluation has its strengths and weaknesses. The strengths include evaluating on two widely used (and publicly available) LTR data sets, comparing against a number of strong baselines, and implementing the approach within the open source TF-Ranking framework. The two weaknesses include the lack of proper statistical significant testing to determine whether the differences in performance observed are 'meaningful' and the lack of tree-based baselines (such as LambdaMART). The missing LambdaMART baseline is the most serious concern that I have, as it is an exceptionally strong baseline that is worthy of comparison and will help determine the actual significance/impact of the proposed method.

Despite all of the positives of this paper (well written, interesting method), I am unable to recommend it for publication as of this time given the concerns with the experimental evaluation (especially the missing LambdaMART baseline).

===

Update based on authors' rebuttal: The rebuttal provided the requested statistical significance tests and the results look favorable for PiRank, thereby providing better support for the authors' claims. I still believe it would be worthwhile to include a comparison against the current state-of-the-art (even if it is not neural-based and even if PiRank does not outperform it -- this is mostly as a point of comparison to provide additional context). I have increased my rating for this paper (from 4 => 6) to reflect this new information.

**Time Spent Reviewing:**

2

---

> ### Author Response · Authors · 2021-08-22
> **Response to reviewer's feedback**
>
> We thank the reviewer for their review and answer the outstanding comments and questions below.
>
> Q:  [...] lack of proper statistical significant testing to determine whether the differences in performance observed are 'meaningful'
>
> A: We have included the statistical significance testing results here: https://sites.google.com/view/neurips-pirank/home#h.4u3rrq8fhr18 (Tables 1 & 2). We followed a methodology similar to that found in a recent work [1]: for each metric and dataset, the best performing method is determined, then a one-sided paired t-test is performed on query-level metrics on the test set to compare the best method with every other method. We show in bold the best performing method as well as all other methods which were not significantly worse than the best one at a 95% significance level. We find that PiRank is amongst the top performing methods in almost all cases and is the single best performing method for a majority of the metrics.
>
> Note that computing the significance test statistics requires query-level statistics; these were not accessible in the TFRanking library. During the rebuttal period, we had to develop a patch to the TFRanking library to address this issue. Further, we noticed that since our earlier experiments, the recommended TFRanking architecture has been updated to a significantly better version [1]. While some of the details are missing, we reproduce it to the best of our understanding and report updated results for all methods.
>
> Q: The missing LambdaMART baseline is the most serious concern that I have, as it is an exceptionally strong baseline that is worthy of comparison and will help determine the actual significance/impact of the proposed method.
>
> A: Indeed, LambdaMART is a very strong baseline as a learning to rank approach. The scope and focus of this work is on differentiable losses that can be used to train neural networks. Our evaluation protocol is the same as [1] in that we benchmark our proposed loss function against baseline loss functions using the same neural network architecture and training setup. This allows us to fairly compare the different losses. Consequently, we built on top of the open source TFRanking framework which does not include LambdaMART (a tree based approach). We will clarify this in the final version and explicitly note that our study is limited to neural network based approaches and other non-neural network based methods like LambdaMART could in principle be better than our approach.
>
> We are very grateful for the reviewer’s helpful feedback and comments. Please let us know if there are any remaining questions; we will be glad to address them!
>
> [1] Reddi, Sashank J. et al. “RankDistil: Knowledge Distillation for Ranking.” AISTATS (2021).

---

### Official Review · Reviewer_b1Fc · 2021-07-19

**Rating:** 6
**Confidence:** 4

**Summary:**

This paper proposes a new class of surrogates called PiRank for ranking metrics optimization. In particular, it uses temperature-controlled relaxation to the permutation matrix to derive a continuous and differentiable sorting operator. Due to the large size of the ranking lists in practical applications, this paper also proposes a divide-and-conquer method to reduce the complexity of metric computation. Experiment results show that the proposed methods can achieve promising results on two popular benchmarks.

**Ethical Concerns:**

None as I perceived.

**Limitations And Societal Impact:**

In the experiment results, only PiRank-NDCG is reported, which is also used under metrics other than NDCG. Does this mean PiRank on other metrics does not work as well?

The divide-and-concur method is proposed to improve the efficiency of PiRank. But will be approximation still correspond to the correct ranking metric? Besides, in experiments, only list size under 100 is studied, which does not show the effectiveness of the divide-and-conqur variant.

**Main Review:**

ORIGINALITY: There are existing works that approximate the ranking metrics so that they can be optimized by gradient based methods. Most of them aim to derive a bound of a certain ranking metric. The key is to smooth the rank calculation. In comparison, PiRank utilizes a permutation matrix to represent the rank as in NeuralSort, so that the rank does not need to be directly calculated based on the predicted scores. Then it applies continuous relaxations on the permutation matrix to smooth the ranking metrics, which looks novel.

QUALITY: The paper proposes PiRank for differentiable ranking metric approximations, which can control the level of approximation based on temperature parameters. PiRank relaxes the permutation matrix to achieve differentiability, which is neat and flexible. A divide and conquer method is also proposed, which completes the loop for practical learning to rank tasks. Experiments are performed on two benchmark datasets against several popular baselines. Overall the paper is complete well finished.

The arguments that other surrogates are loosely related to ranking metrics or simple smoothed versions need to the further investigated. Most surrogates are deriving bounds of the ranking metrics, which are directly related to the metrics, though the bounds may have different levels of looseness. It would be interesting to show that PiRank is a better approximation of the metrics under useful temperatures.

In lines 79-84, pairwise approaches have been proved to be able to approximate the metrics on the full list when aggregated over all pairs (see Wang et al. 2018, The lambdaloss framework for ranking metric optimization.).


CLARITY: The paper is clearly presented and easy to follow. However, the advantage of PiRank over other surrogates could be further elaborated. For example, PiRank also needs to approximate ranking metrics, which is doing the same thing as others. Though PiRank can achieve the exact metric when the temperature is zero, it is useless in this case.

SIGNIFICANCE: Ranking metrics approximation and optimization is an important field in learning to rank. PiRank provides a new perspective in doing this task. The proposed method is neat and easy to apply in practice.

I have read the authors' response and decided to keep my rating of the paper.

**Time Spent Reviewing:**

3

---

> ### Author Response · Authors · 2021-08-22
> **Response to reviewer's feedback**
>
> We thank the reviewer for their review and answer the outstanding comments and questions below.
>
> Q: In the experiment results, only PiRank-NDCG is reported, which is also used under metrics other than NDCG. Does this mean PiRank on other metrics does not work as well?
>
> A: We focused on NDCG as it is the most prevalent list-based metric in the literature. Our benchmark experiments  (Table 1) also showcase that NDCG-based PiRank is the best performing metric on ARP for the benchmark datasets. If ARP is the desired metric of interest, then we expect PiRank-ARP to be a better surrogate.
>
> Q: The divide-and-conquer method is proposed to improve the efficiency of PiRank. But will the approximation still correspond to the correct ranking metric? Besides, in experiments, only list size under 100 is studied, which does not show the effectiveness of the divide-and-conquer variant.
>
> A: Yes, the loss function based on the divide-and-conquer method is an approximation of the ranking metric at finite temperatures but recovers the metric exactly in the limit. See Proposition 1 in Appendix B for a formal result.
> We have synthetic experiments in Section 4.2 which show the computational effectiveness of the divide-and-conquer approach, see especially Figure 7. For our benchmarks on publicly available real world datasets, the list size is low and hence, we did not need any more computational optimizations based on the divide-and-conquer approach. Note, however, this is not the case for industrial applications (eg, web search engines) where it is common to have very large list sizes.
>
> Again, we are very thankful for the reviewer’s helpful feedback and comments. Please let us know if there are any remaining questions; we will be glad to address them!

---

### Official Review · Reviewer_WcGQ · 2021-07-20

**Rating:** 7
**Confidence:** 3

**Summary:**

This paper tackles the problem of Learning To Rank. The idea is to relax ranking metrics (eg DCG), that are non differentiable with respect to the model parameters theta. Originally, these metrics measure how well the predicted ranking (induced by the scores predicted by the model) places high items with high true relevance labels. The relaxed metrics replace the permutation matrix in the original metrics by a relaxed, unimodal matrix, parametrized by a temperature.

This relaxation was proposed in Neuralsort, but is of quadratic complexity in the total number of items L at each time. In this paper, the authors propose a computationally efficient version of Neuralsort, truncated at the k first items, relying on a divide and conquer strategy. This approach is of much lower complexity, expressed in terms of L,k, and d the depth of the tree.

Finally, the method is evaluated on benchmark learning to rank datasets. It obtains similar or better performance than sota methods.

**Ethics Review Area:**

["I don’t know"]

**Limitations And Societal Impact:**

The authors have adequately addressed the limitations and potential negative societal impact of their work.

**Main Review:**

Pros:
1. The paper is very clearly written, I enjoyed reading it.

2. The proposed method seems very attractive:
     - it is very interesting computationally when L is large at train time when using trees of depth d>1, and performs similarly or better than widely used competitive methods
- as the temperature goes to zero, we recover the true, non differentiable loss. This contrasts with previous listwise approaches that consider surrogate losses that may be poorer approximations of the objective.

Cons - comments/questions about the hyperparameters - :
1. I was a bit surprised to read in Appendix C “	.	Parameters specific to  NeuralSort and PiRank are tau = 5 and d = 1 across all experiments except for ablation “. If d=1, if I understood well, there is no real computational benefit (comparison of the complexity given l214 and L^2)
2. How is the temperature chosen, e.g. in Table 1? I see in Appendix C that it was fixed to 1. The authors seem to have investigated the effect of different temperatures in sec 4.2; but if I understood well, the temperature is fixed during the training. Did you try an evolving temperature, such as in simulated annealing? e.g. tau goes from 1 to 0 along iterations. If yes, was it the best choice compared to a fixed tau?
    3.  When tau is large, it is not clear to what extent the surrogate loss you propose is better than the ones considered in the literature.


Globally, I remain positive about the contributions of this paper.

**Time Spent Reviewing:**

5

---

> ### Author Response · Authors · 2021-08-22
> **Response to reviewer's feedback**
>
> We thank the reviewer for their review and answer the outstanding questions below.
>
> Q: If d=1, if I understood well, there is no real computational benefit (comparison of the complexity given l214 and L^2)
>
> A: That is correct. The introduced loss allows a trade-off between scaling and accuracy with the parameter d. Setting d>1, the complexity decreases (L214) but this could potentially reduce the accuracy due to the truncations of intermediate results in the merge sort (L189). As the public open-source datasets can fit in memory for d=1, we used this setting to optimize for accuracy. Values d>1 should be used when applied on datasets with larger lists that would not fit in memory otherwise, as is common in industrial use cases e.g., web search. To demonstrate the computational benefits of the divide and conquer approach, we include evaluations on a synthetic dataset (L272-282).
>
> Q: Did you try an evolving temperature, such as in simulated annealing? e.g. tau goes from 1 to 0 along iterations. If yes, was it the best choice compared to a fixed tau?
>
> A: We tried an annealing schedule that follows a multiplicative decay schedule over 100 epochs. We did not see much difference with this schedule; in fact, the optimization was relatively unstable. The reviewer can refer to the Figure in https://sites.google.com/view/neurips-pirank/home#h.hg5xvwmqeqib for the performance curves. We agree that the design of better annealing schedules is indeed a promising direction for future study!
>
> Q: When tau is large, it is not clear to what extent the surrogate loss you propose is better than the ones considered in the literature.
>
> A: Indeed, when tau is large, we find that the surrogate loss is not a desirable objective for optimization because the approximation quality is very poor. The reviewer can refer to the Figure on https://sites.google.com/view/neurips-pirank/home#h.hg5xvwmqeqib for an example with very high temperature. In terms of the bias-variance tradeoff, this corresponds to the scenario of low variance, but high bias in the estimated gradients.
>
> Again, we are very thankful for the reviewer’s helpful feedback and comments. Please let us know if there are any remaining questions; we will be glad to address them!

---

### Decision · Program_Chairs · 2021-09-27

**Decision:**

Accept (Poster)

**Comment:**

This work proposes a new method for approximation of sorting and ranking operations in a differentiable manner, with a focus on the scalability of the operations.

The reviewers were unanimous that this paper should be accepted, and I agree with them. Since the problem and methodology cannot be considered very novel (many similar methods exist), and the numerical results are not a major breakthrough, I am not recommending a spotlight.